# Dengue and chikungunya virus loads in the mosquito *Aedes aegypti* are determined by distinct genetic architectures

**Mario Novelo**[1,2], **Heverton LC Dutra**[2,3], **Hillery C. Metz**[1,2], **Matthew J. Jones**[2,3], **Leah T. Sigle**[1,2], **Francesca D. Frentiu**[4], **Scott L. Allen**[5], **Stephen F. Chenoweth**[5], **Elizabeth A. McGraw** [2,3]*

**1** Department of Entomology, The Pennsylvania State University, University Park, Pennsylvania, United States of America, **2** Center for Infectious Disease Dynamics, The Huck Institutes of the Life Sciences, The Pennsylvania State University, University Park, Pennsylvania, United States of America, **3** Department of Biology, The Pennsylvania State University, University Park, Pennsylvania, United States of America, **4** Centre for Immunology and Infection Control, School of Biomedical Sciences, Queensland University of Technology, Herston, Queensland, Australia, **5** School of Biological Sciences, The University of Queensland, St. Lucia, Queensland, Australia

* eam7@psu.edu

**Data Availability Statement:** All viral load data, RNAseq read count data, and BAM files are available through figshare DOI 10.6084/m9. figshare.20483691.

## Abstract

*Aedes aegypti* is the primary vector of the arboviruses dengue (DENV) and chikungunya (CHIKV). These viruses exhibit key differences in their vector interactions, the latter moving more quicky through the mosquito and triggering fewer standard antiviral pathways. As the global footprint of CHIKV continues to expand, we seek to better understand the mosquito's natural response to CHIKV—both to compare it to DENV:vector coevolutionary history and to identify potential targets in the mosquito for genetic modification. We used a modified full-sibling design to estimate the contribution of mosquito genetic variation to viral loads of both DENV and CHIKV. Heritabilities were significant, but higher for DENV (40%) than CHIKV (18%). Interestingly, there was no genetic correlation between DENV and CHIKV loads between siblings. These data suggest *Ae. aegypti* mosquitoes respond to the two viruses using distinct genetic mechanisms. We also examined genome-wide patterns of gene expression between High and Low CHIKV families representing the phenotypic extremes of viral load. Using RNAseq, we identified only two loci that consistently differentiated High and Low families: a long non-coding RNA that has been identified in mosquito screens post-infection and a distant member of a family of Salivary Gland Specific (SGS) genes. Interestingly, the latter gene is also associated with horizontal gene transfer between mosquitoes and the endosymbiotic bacterium *Wolbachia*. This work is the first to link the SGS gene to a mosquito phenotype. Understanding the molecular details of how this gene contributes to viral control in mosquitoes may, therefore, also shed light on its role in *Wolbachia*.

**Funding:** The work was supported by an NIH R01 grant to SFC and EAM (AI143758). The funders had no role in study design, data collection and analysis, decision to publish, or preparation of the manuscript.

**Competing interests:** The authors declare no competing interests.

## Author summary

The virus chikungunya (CHIKV) that causes long term arthritis symptoms in humans is transmitted to through the bite of the *Aedes aegypti* mosquito. CHIKV, for which there is no vaccine, is becoming increasingly common across the globe. We therefore need to understand the mosquito's own ability to control CHIKV, as we may use that knowledge to create resistant mosquitoes through genetic modification. We show that the mosquito has very little ability to respond genetically to CHIKV, indicating low potential for the mosquito to evolve resistance. We also found that the genetic basis of CHIKV viral loads appears distinct from dengue, another common virus. As such, any strategy for engineering virus-resistant mosquitoes may need to be virus-specific or focus on the few overlapping genes in the mosquito response. Last, when we examined the mosquito genes whose expression differed between high and low-load virus lineages, we discovered a gene that was highly expressed in low-load families and therefore, potentially acting as a virus controller. Interestingly, a homolog of this gene has been discovered in the genome of the *Wolbachia* endosymbiont, itself known to limit virus replication inside its insect hosts. The functional importance of this homolog in virus control should therefore be explored in both mosquitoes and *Wolbachia*.

## Introduction

Arthropod-borne pathogens account for more than 17% of all infectious diseases globally [1]. Among the most prevalent are dengue (DENV) and chikungunya (CHIKV) viruses, with over half of the world's population at risk. There are no effective vaccines or treatments for either pathogen [2] and morbidity can be severe—with some cases leading to dengue shock syndrome [3], or neurological conditions [4], and debilitating, chronic arthralgia [5]. Both viruses inflict significant socioeconomic harm as well.

Both DENV and CHIKV are single-stranded positive-sense RNA viruses belonging to the *Flaviviridae* and *Togaviridae* viral families, respectively, and both are transmitted primarily through the bites of *Aedes aegypti* mosquitoes [6,7]. However, the two pathogens have distinct histories. DENV has a long association with *Ae. aegypti*: virus and vector together becoming ubiquitous across the tropics [8,9] with descriptions of DENV-like illness (aka 'break bone fever') appearing as early as 1635 in the Americas [10]. In contrast, CHIKV has remained a relatively rare pathogen, causing only small outbreaks on some islands of the Indian Ocean until recently [11]. The two viruses have thus had unequal opportunities to coevolve with *Ae. aegypti*, their shared primary mosquito host. However, CHIKV cases have recently exploded, with extensive range expansion in some areas of the world [12]. The virus was first reported in the Americas in 2013, and by 2018 there had been more than 3 million cases in 45 countries [13,14]. Despite this sharp rise in CHIKV infections around the globe, our understanding of how this pathogen infects and interacts with *Ae. aegypti* remains incomplete.

Arboviral infection in the mosquito is a complex and dynamic process shaped by the genetic variation of both the mosquito and the virus [15]. Natural *Ae. aegypti* populations can vary in their susceptibility to arboviruses, including DENV and CHIKV [16–20], yet diverse populations from across the tropics are highly susceptible to CHIKV infection [20–24]. For example, *Ae. aegypti* populations from the Americas, where CHIKV appeared less than a decade ago, transmitted CHIKV in their saliva at rates as high as 83% [24]. Moreover, CHIKV infectious particles have been detected in *Ae. aegypti* salivary glands as early as 2 days post-infection (DPI), indicating that the dissemination of the virus within the mosquito is rapid

[25]. In contrast, DENV particles are not detected in salivary glands until later [26,27]. Such differences may stem from variation in viral replication inside the mosquito, where both mosquito genetic factors and specific pathogen infection mechanisms shape transmission patterns.

After ingesting a virus-laden blood meal, molecular interactions between mosquito cells and the pathogen begin. First, viral particles attach to cellular receptors, initiating endocytosis or activating specific pathways to access cells [26]. How DENV and CHIKV enter mosquito cells is not completely understood, though the two arboviruses are thought to differ. For DENV, the specific identity of mosquito cellular receptors remains elusive, although several putative receptors have been suggested [27,28]. One of these, *Prohibitin*, is a ubiquitously expressed, conserved protein shown to interact with DENV-2 [29], and there is mixed evidence for a role for clathrin [30,31]. CHIKV cellular internalization is thought to be mediated through clathrin-mediated endocytosis (CME), a key process in vesicular trafficking [32–34]. Both viruses are icosahedral in shape, however, CHIKV presents a T = 4 geometry while DENV has a pseudo T = 3. CHIKV genome is organized with 4 non-structural proteins at the 5' followed by 4 structural proteins, while DENV genome organization is inverted, with the structural proteins at the 5' end followed by non-structural proteins. Upon entry, both viruses are uncoated through intracellular specific conditions, such as acidic pH, and replicate in the cytoplasm [35]. Viral RNA is then translated into a polyprotein (or polyproteins in the case of CHIKV [36]) which is processed during maturation by host and viral enzymes. Glycoproteins are then inserted in the endoplasmic reticulum, where virion assembly occurs. For DENV, immature virions are further processed through the Golgi membrane system by carbohydrate addition and modification and then follow the exocytosis pathway and are released into the extracellular space by fusion of virion-containing vesicles with the plasma membrane. CHIKV is post transcriptionally modified and glycosylated in the endoplasmic reticulum and translocated to the Golgi apparatus to be packed in vesicles and delivered to the cell membrane. Here, further maturation occurs by acquisition of membrane envelope, then virions are released via exocytosis. In sum, there are both similarities and differences in how the two viruses gain entry to cells and use cellular pathways to replicate [37,38].

Cellular tropism for both DENV and CHIKV appears widespread to multiple mosquito organs, including midgut, salivary glands, fat body tissue, nervous, tracheal, and reproductive systems [39–45]. Interestingly, variation in the replication rates between of the two viruses inside *Ae. aegypti* have been reported. Specifically, a study measuring viral RNA levels found CHIKV had higher copy number than DENV in midguts, and CHIKV was also detected far earlier than DENV in salivary glands [46]. Moreover, co-infection with DENV and CHIKV increases overall mosquito infection rates up to 100% for both viruses, while also enhancing CHIKV and DENV viral replication in the midgut and salivary glands, compared to each virus alone [46]. These findings suggest the cellular and molecular mechanisms by which *Ae. aegypti* responds to DENV and CHIKV infection may interact or overlap, though enhancement during co-infection may also arise from energetic constraints.

Viral infections trigger cellular and humoral immune responses inside the mosquito. The core pathways of the insect immune response are the Janus Kinase-Signal Transducer Activator of transcription (JAK/STAT), Toll, Immune Deficiency (IMD) and RNA interference (RNAi) [26]. Viral components also interact with pattern recognition receptors, triggering processes such as melanization and the production of anti-microbial peptides (AMP), including attacin, defensin and cecropin, among others [47–49]. Interestingly, some of the traditional innate immune pathways (Toll, JAK/STAT, IMD) that have been shown to contribute to the reduction of DENV replication both *in vitro* and *in vivo* play little role in limiting CHIKV infection. Specifically, exogenous activation of the JAK-STAT pathway has been shown to modulate DENV infection but did not enhance resistance to CHIKV [50]. Moreover, CHIKV

infection significantly represses the Toll pathway, limiting its efficacy, and the IMD pathway also does not mediate CHIKV infections [51]. Of the major insect immune pathways known to control infection of arboviruses such as DENV, only the RNAi pathway has been shown to play a vital role in limiting CHIKV replication. Specifically, knockdown of Argonaute2 (*AGO2*), an RNAi effector molecule, resulted in significant increase of viral RNA replication and titers [51]. Consistent with this evidence that mosquito responses to the two viruses are distinct, transcriptomic profiles of DENV and CHIKV infected mosquitoes revealed little overlap between responsive genes [52].

One path to resolving these heterogenous findings is to exploit natural genetic variation in infection response in *Ae. aegypti*. By examining population level genetic variation in antiviral response, several recent studies have discovered novel, non-canonical antiviral genes in Drosophila [53,54] and in *Ae. aegypti* [55]—demonstrating the promise of this approach. Here, we used a modified full-sib breeding design to assess the family-level variation in CHIKV and DENV loads, using a Mexican population of *Ae. aegypti*. By examining viral loads in a common set of mosquito families, our design can uncover the contribution of genetic variation to viral load variance as well as assess the extent to which mosquito responses to these viruses are due to shared or independent genetic mechanisms. We also examined transcriptional differences in mosquito families representing the phenotypic extremes for CHIKV load (lowest and highest mean total body loads), allowing us to identify individual genes that may underlie this CHIKV variation. Our work thus provides insights into the basis of the vector's own natural immune response to CHIKV and DENV, and points to specific candidate antiviral genes to target with emerging insect genetic tools, such as CRISPR-Cas9 [56,57].

## Results

### Family-level variation in the heritability of DENV and CHIKV viral loads

To estimate the heritability of DENV and CHIKV loads, we examined viral loads in whole bodies of female mosquitoes previously fed either virus (Fig 1, experiment 1). The data came from mosquitoes representing 37 mosquito families bred within a modified full-sib breeding design. After starting with over 600 individual mating pair crosses, we obtained data for 37 because most families either did not produce enough eggs or blood feed sufficiently. The siblings in each family were split in half, and each set was fed one virus, allowing us to test for genetic correlation. Mosquitoes were assayed for viral loads 7 days post-infection by extracting RNA and performing DENV or CHIKV-specific RT-qPCR on 3–5 individuals per family. Viral loads were extremely variable across families, spanning from hundreds to millions of copies; $10^2$–$10^7$ per body for DENV and $10^2$–$10^8$ per body for CHIKV. Both $H^2$ values were significantly greater than zero, indicating a genetic basis underlies variation in viral loads for both DENV and CHIKV. The broad-sense heritability ($H^2$) of DENV load was estimated to be 0.40 (Fig 2A. LRT: $X^2$ = 24.8, $P<0.001$). For CHIKV load, $H^2$ was 0.18 (Fig 2B. LRT: $X^2$ = 5.9, $P$ = 0.015). Both viral load distributions exhibit some bimodality, a common feature of infection in mosquitoes we have explored previously [15].

### Genetic correlation between DENV and CHIKV across mosquito families

We estimated the genetic correlation, $r_{DENV,CHIKV}$, to assess the extent to which genetic variation underlying the control of DENV and CHIKV loads was shared. A value indistinguishable from 1 would indicate they are effectively the same genetic 'trait' whereas values closer to zero indicate that the traits are genetically independent. We estimated $r_{DENV,CHIKV}$ = 0.1 and this value was significantly less than 1.0 (LRT: $\chi^2$ = 5.7, df = 1, $P$ = 0.017) but could not be distinguished from zero (LRT: $\chi^2$ = 0.1, df = 1, $P$ = 0.752). We found no evidence of genetic

## Experiment 1

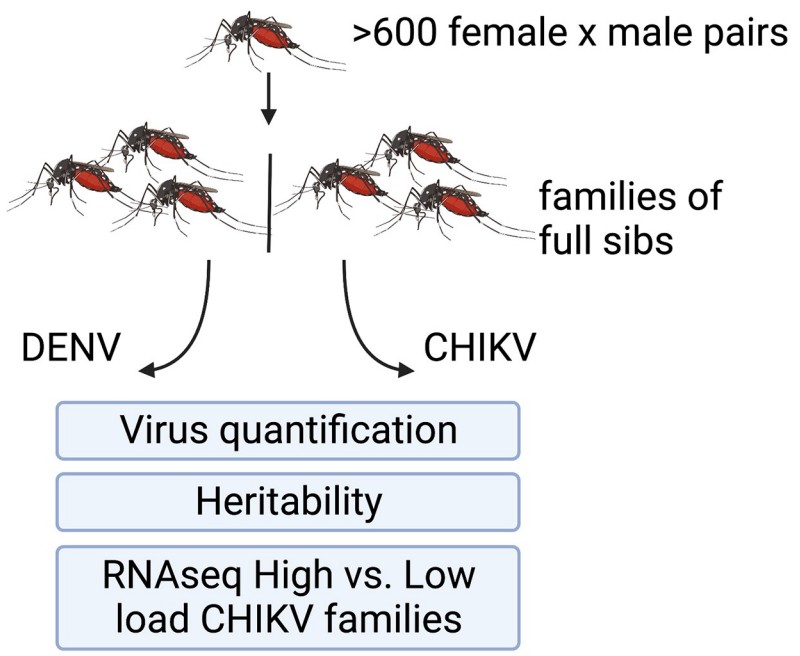

## Experiment 2

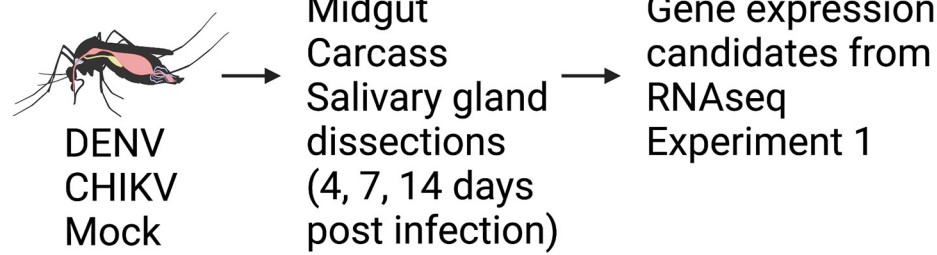

**Fig 1. Overview of experimental design.** In experiment 1, we studied CHIKV and DENV loads in siblings from single female x male pair crosses and calculated the heritabilities and genetic correlations of the loads. We then carried out RNAseq comparing two groups; families with the four highest and four lowest mean CHIKV loads. In experiment 2, we reared a separate set of mosquitoes from the same population and exposed them to CHIKV, DENV, or mock (uninfected cell culture control) infection through blood-feeding treatments and then dissected key tissues (midgut, carcass, salivary glands) at three timepoints post-infection (4, 7, 14). We then examined patterns of gene expression of the top candidate gene from the RNAseq analysis in experiment 1 across treatments and tissues x time. Created with BioRender.com.

correlation between DENV and CHIKV loads between siblings of the same family based on our analysis of 37 families (Fig 3. Pearson's $r$ = 0.0018, $P$ = 0.99). Furthermore, the correlation was not significantly greater than zero (LRT: $X^2$ = >0.001, $P$ = 1.00). This is underscored by the lack of any overlap between the five families representing the extremes of the distributions for both viruses (Fig 2). Specifically, high-load families for DENV were families 77, 18, 95, 1, and 32, while the same for CHIKV were families 59, 26, 81, 12, and 95. The family numbers were also different for the five low-load families. The independence between viral load traits for siblings indicates a lack of shared control mechanisms for the two viruses.

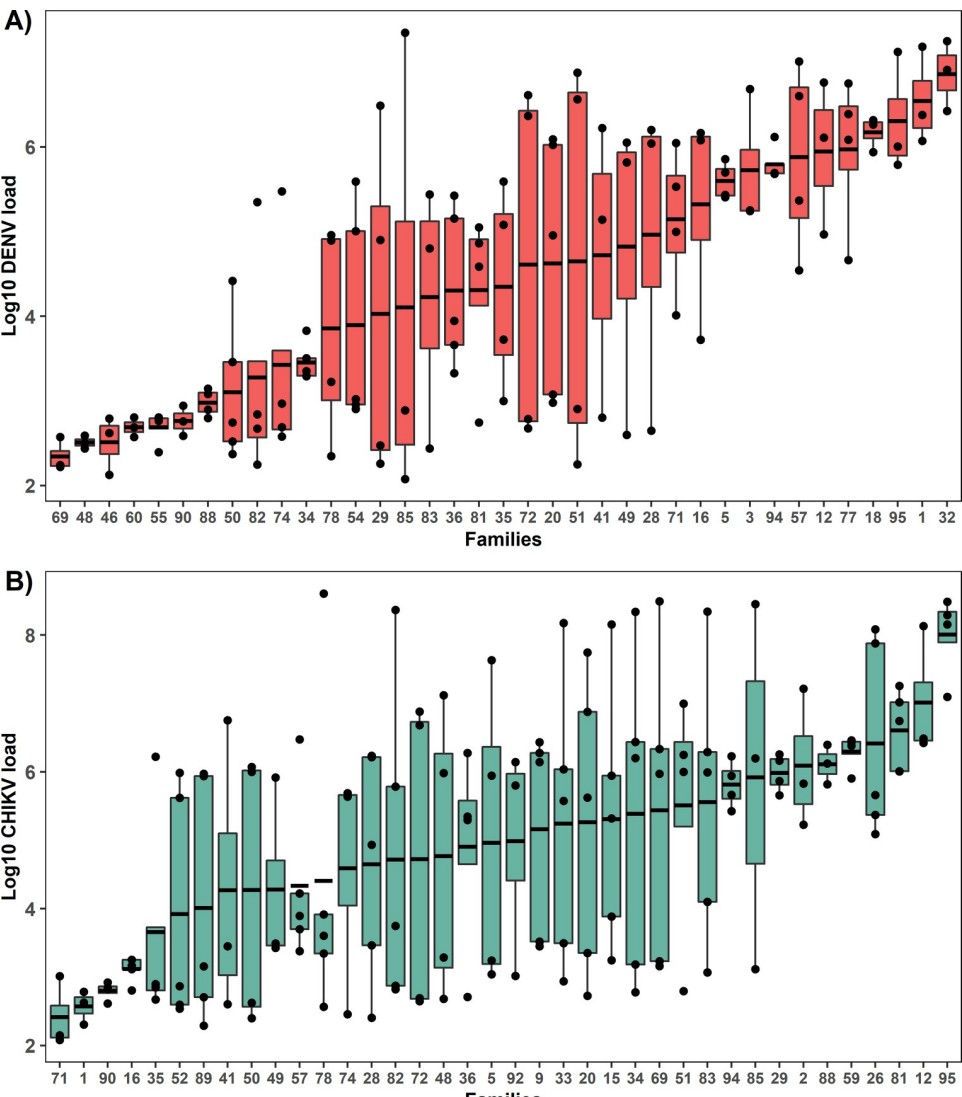

**Fig 2. Ranked families by viral load.** We ultimately obtained data from 37 families from >600 initial paired crosses.
**A)** DENV load-ranked families and **B)** CHIKV load-ranked families. Each point represents a single infected mosquito.
Box plots show mean viral load ± SEM of each distinct *Ae. aegypti* family. Whole mosquito viral load was log10
transformed. N = 3–5 individuals per family.

## Gene expression of families representing the phenotypic extremes of viral loads for both DENV and CHIKV

We then tested whether the mean viral loads differed between each virus's four lowest and
four highest families. For both DENV ($w = 256$, df = 1, $P<0.0001$) and CHIKV ($w = 393$,
df = 1, $P<0.0001$), families at the phenotypic extremes differed. Because there has already been
substantial profiling of gene expression for *Ae. aegypti* with differences in susceptibility and
viral loads for DENV (reviewed in [58]), we focused on understanding if there were clear dif-
ferences in gene expression between high and low families for the less well-studied CHIKV.
We obtained RNA-Seq expression data for 10,014 genes in the families, each from the high
and low family categories (L: 1, 16, 71, 90, H:12, 26, 59, 81). Following multiple test correc-
tions, we identified only two loci that differed consistently (adjusted $P<0.05$) between the

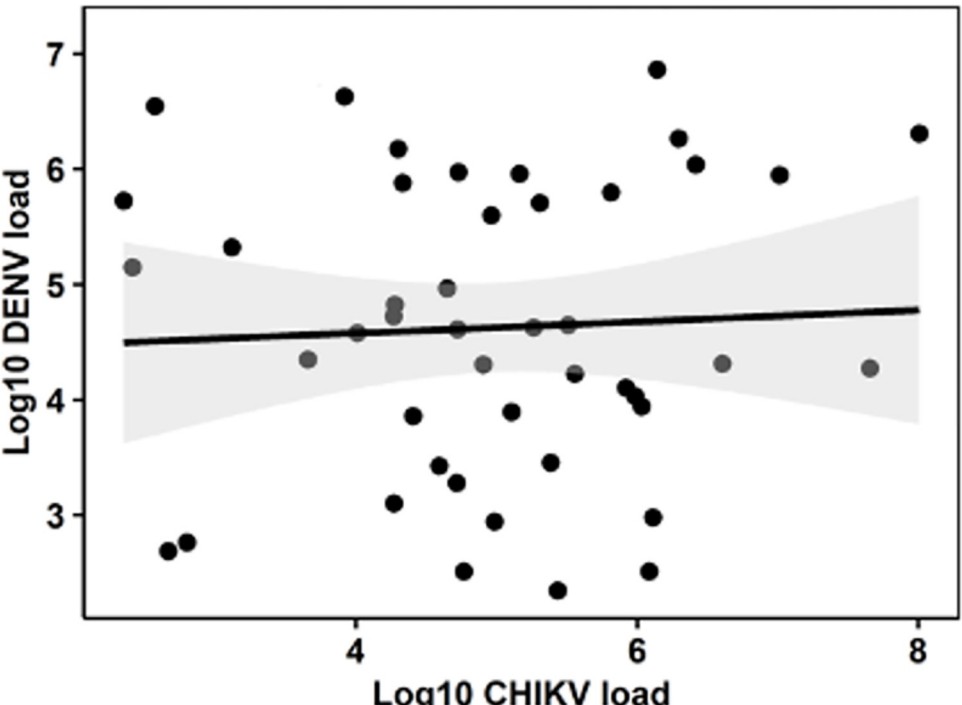

**Fig 3. Scatter plot showing the relationship between mean DENV and CHIKV loads from individual families.**
Gray area represents the 95% confidence interval of the regression line. N = 37 mosquito families.

families representing the extremes (Fig 4). The first, LOC110676965, corresponds to
AAEL026917, encoding a Zinc finger-containing protein that was identified previously as
upregulated in an RNAseq dataset for dengue-infected *Ae. aegypti* [59]. Similarly, this same
gene's expression responds to Zika infection [60]. The second locus, LOC5564245, encodes
AAEL004181-PA, an uncharacterized protein. AAEL004181-PA has no previously reported
phenotypes in mosquitoes but is a distant relative in a family of SGS or salivary gland-specific
proteins [61]. Interestingly, this locus has been identified as a horizontally transferred gene
with the insect endosymbiont, *Wolbachia* [61–63]. This explains why the next six hits resulting
from a protein blast search [64] identify genes in several strains of *Wolbachia* from *Culex* spe-
cies, *Drosophila melanogaster*, and *Drosophila yakuba*. Similarities range from 77 to 99%.
While there is some controversy over the direction of the horizontal transfer, the preponder-
ance of the evidence is that AAEL004181 is fundamentally a mosquito gene that was trans-
ferred into *Wolbachia* multiple times [61,63]. In *Wolbachia*, there are two genes, positioned
side by side in the genome (known as loci WD0512 and WD0513 in the *w*Mel genome [62],
that together are homologous to the mosquito gene. Both WD0512 and WD0513 are actively
transcribed [65], suggesting they are functional in *Wolbachia*.

Despite multiple attempts to design primers for the LOC110676965, we were unable to find
a set that amplified the locus. This could have resulted from a poor understanding of the
underlying intronic structure or, more likely, simply very low-level expression. In our high
CHIKV families, for example, the RNAseq read counts were very low (median 0.92 reads/sam-
ple) compared to the low families (median 80.7). We were able to design primers to amplify
AAEL004181 that detected expression in our samples. We then carried out a separate and
more detailed map of expression (Fig 1, experiment 2) over time and across several mosquito
tissues (without family structure) to understand its responsiveness to infection. During

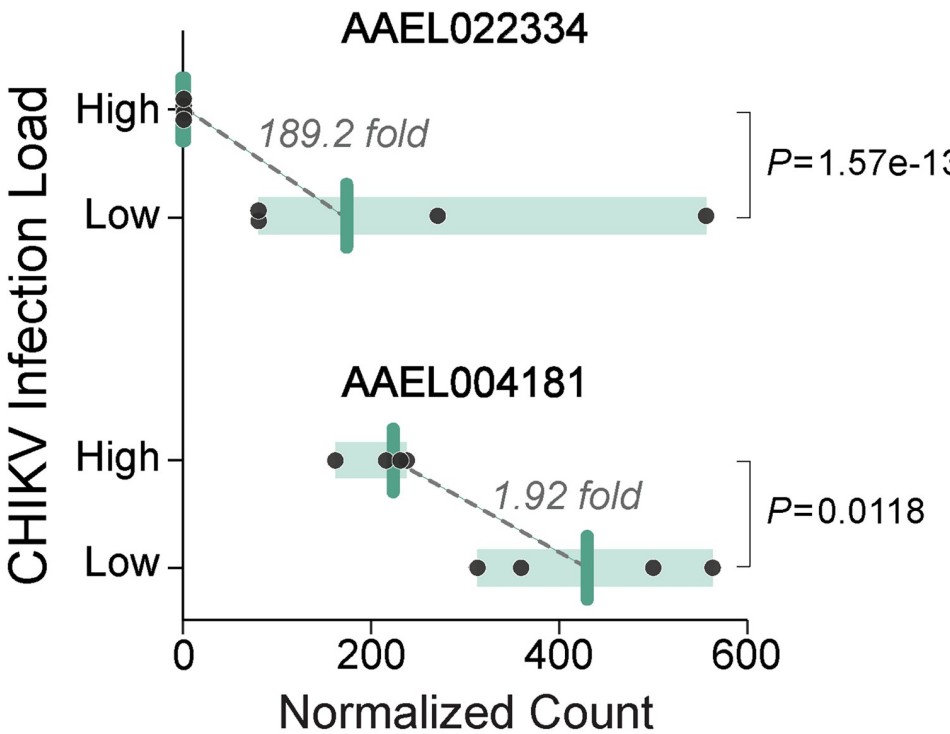

**Fig 4. RNAseq counts for the two loci with significant expression differences between high and low CHIKV-load families.** Points depict normalized read counts of individual families, bars show group medians, and shaded area demarcates the range for each group. Significance was evaluated using Wald tests while correcting for multiple comparisons.

CHIKV infection, expression of AAEL004181 rises early in the midgut compared to virus-free blood-fed controls (4 dpi, Mann-Whitney U = 55, $P<0.0001$), but then declines with time in the midgut (7 dpi, Mann-Whitney U = 78, $P = 0.0012$) and other tissues (carcass, salivary glands, see Fig 5A for full statistical results). A similar pattern is seen for DENV-infected mosquitoes (Fig 5B).

## Discussion

We performed a modified full-sib breeding design to study the contribution of the mosquito's genetic variation while limiting environmental variation, to loads for each virus. By exploring loads for the viruses in siblings of the same family, we were also able to measure the genetic correlation between the individual loads. We used the distribution of loads to select families representing the phenotypic extremes for CHIKV and tested whether there were genes whose expression differed. Our results are consistent with a polygenic basis for both DENV and CHIKV viral loads. Additionally, the lack of genetic correlation between DENV and CHIKV viral loads across siblings demonstrated the complete independence of the genetic response to these two viruses. Finally, our RNAseq comparisons identified only two genes whose expression differed consistently between high and low CHIKV load families.

The heritability values for viral susceptibility for both DENV and CHIKV were significant but fell within different ranges. For DENV, the $H^2$ was 40%. This is in accordance with previous studies examining susceptibility in *Ae. aegypti* using multiple tissues as proxy. Both Bosio *et al.* [66] and Ye *et al.* [67] reported heritabilities of ~40% in the midgut, head, and saliva of

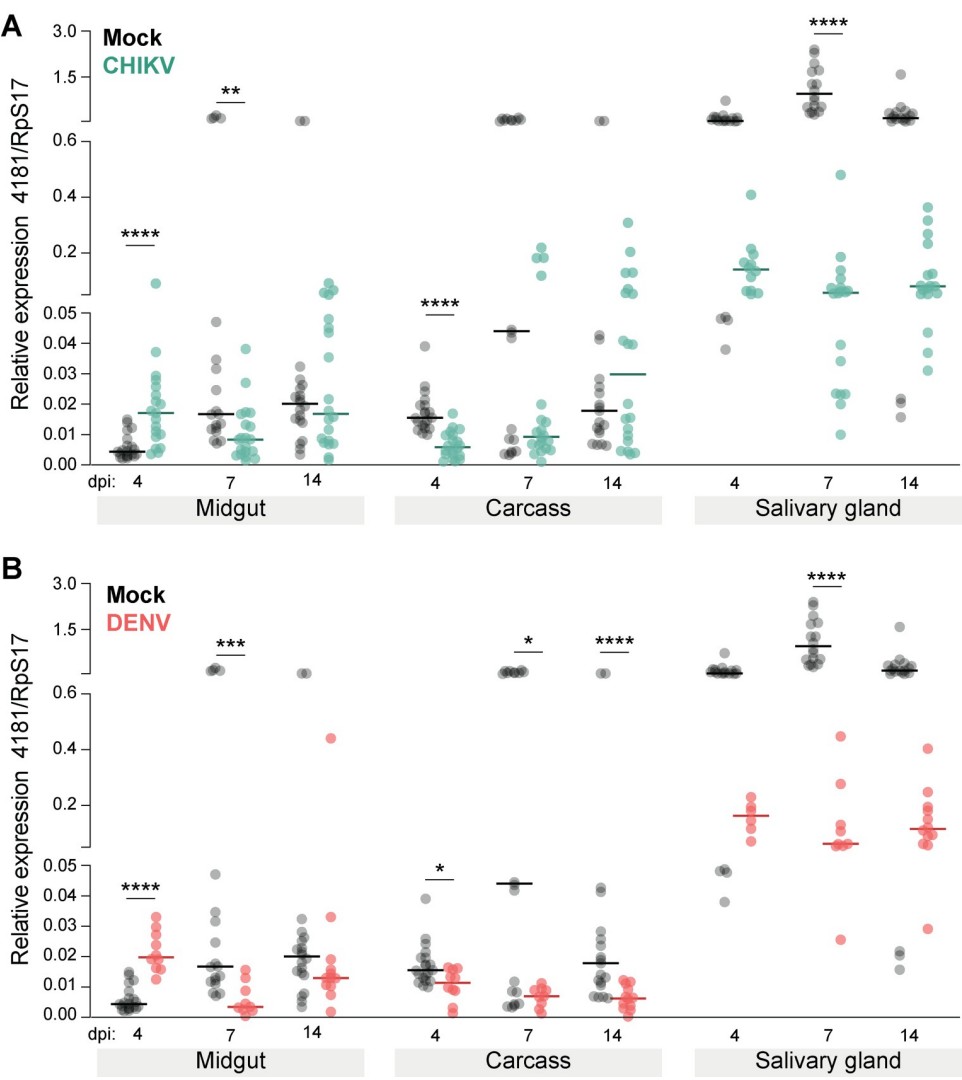

**Fig 5.** Tissue-specific expression of AAEL004181 across time in (A) CHIKV- and (B) DENV-infected mosquitoes compared to expression in mock controls. Points depict relative expression level of AAEL004181 in individual samples, while horizontal lines mark group medians. Significance was evaluated by Mann-Whitney U tests between mock and control. * $P<0.05$, ** $P<0.01$, *** $P<0.001$, **** $P<0.0001$. Note that a day 7 salivary gland outlier in the mock treatment group has been omitted from both panels (value = 7.618). dpi, days post-infection. n = 20 per treatment.

DENV-infected mosquitoes. Taken together, this suggests that for DENV infection load, mosquito genetic variation contributes to vector competence almost as much as all other effects combined, including environmental effects. In our experiment, the heritability of CHIKV load was 18%, with almost half of that estimated for siblings in the same family. The lower heritability for CHIKV suggests that environmental or stochastic factors, as well as viral genotype, have a greater role than the mosquito's own genetic factors in determining variation in load, at least in the tested population [68–70]. CHIKV-infected mosquitoes also exhibited a wider range of viral loads and reached higher absolute viral loads than DENV-infected mosquitoes. The wider range of viral loads in CHIKV was a key contributor to the overall low heritability for this trait. The genetic correlation between siblings for the DENV and CHIKV load trait was not significantly different from zero. This shows independence between the genes that control

load for the two viruses. In general, the primary source of genetic correlation is pleiotropy, though linkage disequilibrium can be a contributing factor [71,72]. Our results, therefore, suggest that the genes controlling CHIKV and DENV viral loads are distinct, unlinked and that they lack pleiotropic effects [73].

Phylogenetic studies on *Ae. aegypti* and DENV have revealed a coevolutionary process between the virus and the co-indigenous mosquito populations [26,47,74] which has in turn shaped susceptibility to the virus. Since CHIKV is an emerging disease, the lack of coevolutionary history with naïve *Ae. aegypti* populations (i.e., in the Americas) could be a source of the differences between the mechanisms that control infection inside the vector [26]. The global spread of CHIKV and DENV was preceded by the global expansion of *Ae. aegypti* and *Ae. albopictus*. While the expansion of *Ae. aegypti* started over 400 years ago [8], the geographic expansion of *Ae. albopictus*, a secondary vector of DENV and CHIKV, started in the 1960's and was previously limited to Southeast Asia [75]. Among the evolutionary forces that drive the evolution of arboviruses, vector interactions may play an important role in selecting viral genotypes [76]. CHIKV emergence in the Indian Ocean was linked to a single mutation that enhanced transmission by *Ae. albopictus* over *Ae. aegypti* [77], leading to the hypothesis that the emergence of arboviruses is partially caused by pathogen adaptations to native vector populations. However, invasive viral genotypes that outcompete local genotypes tend to have enhanced viral replication in vectors [78]. These studies highlight how both mosquito and pathogen genetic variation—and their coevolution—can interact to shape the course of pathogen emergence. Such effects are expected to differ between DENV and CHIKV due to their distinct histories with *Ae. aegypti*.

For a virus such as CHIKV or DENV to act as a selective agent on mosquitoes, it must induce a fitness cost—either directly (i.e., mortality), or indirectly (e.g., energetically, due to immune or stress responses). Fitness changes associated with viral infection in *Ae. aegypti* have been previously reported. Specifically, CHIKV infection in *Aedes* mosquitoes decreased the survival of adults and eggs [79] and disrupted the transcription of genes related to the gonotrophic cycle [80]. Infection with another alphavirus, Mayaro virus (MAYV), reduced fecundity without altering longevity in *Ae. aegypti* [81]. DENV infection in *Ae. aegypti* has been shown to influence feeding behavior and to reduce longevity, fecundity, and oviposition success [82,83]. These and other fitness costs imposed by infection can act as evolutionary forces and may have shaped vector-virus interactions, though they are potentially limited by low infection rates in wild populations: ~1% for DENV [84–86] and ~5–6% for CHIKV [87,88]. Over time, fitness costs related to infection may have contributed to the observed differences in DENV and CHIKV viral loads in *Ae. aegypti*, as the two viruses have very different histories of association with this vector.

A more constant evolutionary force that mosquitoes face and that might have potentially shaped the evolution of insect immunity and susceptibility against medically important arboviruses are insect-specific viruses (ISVs), particularly mosquito-specific viruses (MSVs), which cause no obvious fitness cost in mosquitoes [89], though they may have in the past. The majority of *Ae. aegypti* MSVs are grouped in the *Flaviviridae* viral family and infect both *Ae. aegypti* and *Ae. albopictus* [90]. Other MSV viral families include *Bunyaviridae*, *Mesonviridae* and *Togaviridae*, the last of which has the fewest known MSVs [91]. Interestingly, many of these viruses cluster closely with medically important arboviruses [92]. There is evidence that they interact with mosquito immune pathways such as the RNAi pathway [93], and it is possible MSVs prime the mosquito's immunity to viruses of public health concern, such as DENV and CHIKV, that are not commonly encountered by mosquitoes in the field. The differences in the heritability estimates and the lack of genetic correlation between DENV and CHIKV viral loads observed in our mosquito populations suggest that this could be the case. Different

MSVs could trigger different immune responses. Since the mosquito has likely encountered more MSVs from the *Flaviviridae* (like DENV) than the *Togaviridae* (like CHIKV) across evolutionary time, their immune response could be more 'tailored' to *Flaviviruses* like DENV, and this could possibly contribute to the overall lower viral loads measured for DENV relative to CHIKV in our experiment.

Interestingly, our RNAseq screen returned a locus that has been heavily studied in *Wolbachia*. The *Wolbachia* genes homologous to AAEL004181, WD0512 and WD0513, both sit within an eight-gene operon called Octomom that underlies striking differences in how strains *w*Mel and *w*MelPop affect their arthropod hosts. *w*MelPop is associated with much higher host mortality but also greater antiviral protection in infected *Drosophila*—effects both explicitly linked to greater Octomom copy number and correlated with higher expression levels of Octomom genes [94–96]. This raises the possibility that this homolog inherently produces an antiviral effect, whether it is expressed from a *Wolbachia* genome or a mosquito one, though the mechanism may be more complex [95]. However, molecular details of how AAEL004181/WD0512-WD0513 act in cells—and ultimately upon viruses—are currently lacking and should be investigated in future work. Octomom-associated sequences have been found in several *Wolbachia* strains, but not all, and a few other endosymbiotic bacteria, and mosquitoes [97]. Interestingly, the WD0512-WD0513 genes were lost out of the *w*MelPop-CLA (cell line adapted) strain adapted through long-term serial passage in *Aedes albopictus* cells [98]. The mosquito homolog belongs to a gene family that encodes receptors mediating the invasion of malaria sporozoites into the mosquito salivary gland [63]. Tools for genetic manipulation are sophisticated in mosquitoes [99] but are not available in *Wolbachia*, limiting research progress. However, our results suggest the study of this gene in mosquitoes could potentially also shed light on its role in *Wolbachia*.

Our experimental design has some caveats that may limit the scope of its overall interpretation. Immune effector expression can vary depending on the time post-infection, and it is possible that the expression levels of specific genes might peak, e.g., immediately after blood feeding, and therefore would be missed by our sampling regime [47,100]. However, given the scale of the family-based design, we could only sample a single time point. The expression results shown here could therefore exclude some genes with time- or tissue-constrained differences and would also miss effects stemming from variation in coding sequence as we focused on expression level. Additionally, these are whole body viral loads. More focused work may wish to focus on disseminated virus as a closer proxy for transmissibility [101].

Quantitative genetic estimates—such as we report here—have real-world importance in vector biology as they can help us predict the strength and impact of new outbreaks as well as infection dynamics between vectors and pathogens. Specifically, the higher heritability of load and more homogeneous viral loads of DENV (compared to CHIKV) indicate that selection for higher or lower viral loads in a mosquito population could occur faster for DENV than for CHIKV [68]. The limited genetic variation for modulating CHIKV infections in *Ae. aegypti* may also shed light on the explosive re-emergence of this virus in naïve populations on the American continent, though it remains unclear if the low heritability of CHIKV load is a cause and/or consequence of this recent explosion. Nevertheless, we find the mechanisms that control viral load in mosquito populations are markedly different between DENV and CHIKV. From an epidemiological perspective, if the genes that control viral load for multiple viruses are different and unlinked, this introduces a challenge to vector control methods aiming to target and modify universal loci that could be used against multiple arboviruses. In this case, more virus-specific gene editing approaches would be necessary [102]. Last, the appearance of a *Wolbachia*-associated horizontally transferred homolog in our candidates that associate with CHIKV load is interesting, particularly given the capacity for *Wolbachia* to limit viral

replication via a trait known as pathogen blocking. Unfortunately, without the ability to genetically modify *Wolbachia*, it is hard to test the functional role of this homolog in the symbiont. However, future studies could take advantage of genetic tools in mosquitoes to better characterize their role in viral infections.

## Methods

### Mosquito line and stock-rearing practices

An $F_3$ *Ae. aegypti* mosquito line based on thousands of eggs collected over multiple months from traps placed over the urban environment of Monterrey, Mexico, was reared in the laboratory for three additional generations (i.e., to $F_6$) to expand in preparation for the experiments. Mosquitoes were reared under standard conditions: 26˚C, 65% relative humidity, and a 12 h light/dark cycle. Larvae were maintained on fish food (Tetramin, Tetra). Adults were fed with 10% sucrose solution *ad libitum*. Mosquitoes were fed human blood (BioIVT, Hicksville, NY, USA) for egg collection using a Hemotek system (Hemotek).

### Breeding design

A modified full–sib breeding design [57,67,68] was carried out on the mosquitoes in combination with DENV and CHIKV infection. After expanding the mosquito line over three generations, $F_6$ eggs were hatched in synchrony and reared at low density (~150 larvae per 3 L of RO water). After pupation, males and females were separated and transferred to $30 \times 30 \times 30$ cm cages at a density of ~250 individuals per cage. Six to seven- day old virgin adult females were blood-fed and then 250 virgin males were then added for mating. To achieve data from ~40 families ultimately, which is sufficient for such a design [67], we had to set up a > 600 blood-fed and mated females given the proportions of females that either do not lay or that would lay insufficient eggs or that blood fed poorly. Such females were placed in small individual housings containing moist filter paper. Egg papers were collected and dried 3–4 days later for short-term storage. Only female lineages that laid > 60 eggs could be used for the downstream design, given that half of the eggs would be male, conservative estimates of 80% blood feeding participation, and the need to have individuals in both DENV and CHIKV treatments. The eggs of each family were hatched separately, and after pupation, females were separated and split into two cups with a minimum of 8 individuals per cup. These females were maintained on sucrose until vector competence experiments.

### Virus culture

All experiments were carried out using the CHIKV strain 20235-St Martin 2013 (NR49901, BEI Resources), that is a member of the Asian genotype [103], and the DENV-2 strain 429557-Mex 2005 (NR12216, BEI Resources), both from Latin American outbreaks. Virus was cultured in C6/36 cells, as previously reported [15,57,67,104]. Briefly, C6/36 *Ae. albopictus* cells were grown in RPMI 1640 media (Life Technologies, Carlsband, CA, USA) and supplemented with 10% heat-inactivated fetal bovine serum (FBS, Life Technologies) containing glutamine and 20 mM HEPES (Sigma-Aldrich, St. Louis, MO, USA). Cells were grown to a confluency of 80% and then independently infected with DENV-2 or CHIKV. Infected culture flasks were incubated at 27˚C. For DENV-2, after 7 days post-infection, supernatant was harvested at a titer of $1.0 \times 10^5$ focus forming units per ml (FFU/ml). For CHIKV, supernatant was harvested at 2 days post-infection at a titer of $8.7 \times 10^5$ and diluted to a final concentration of $1.0 \times 10^5$ FFU/ml. This starting infectious dose was intentionally chosen to be low enough to capture variation in mosquito viral loads at 5 days post-infection, particularly for CHIKV that

replicates quickly [25]. Both viruses were stored at -80˚C in 1 ml single-use aliquots for vector competence experiments.

## Mosquito infections

Methods for oral mosquito infections have been fully described previously [15,57,105]. Prior to infections, mosquitoes were starved for 24 h. Half of the 6–7-day old females of each family were challenged with DENV-2 (1st cup) and the other half (2nd cup) with CHIKV at equal viral titers using a 1:1 mix of the frozen titrated aliquots and human blood. Mosquitoes were allowed to feed for 30 minutes. Shortly after, they were anesthetized with $CO_2$ and unfed individuals were removed and discarded. Feeding rates were very high, 80–90% for both DENV and CHIKV-laden blood. All CHIKV work was carried out at the Penn State Eva J. Pell ABSL-3 laboratory and DENV work in the McGraw BSL-2 lab.

## Viral quantification

Seven days post-infection, whole mosquitoes were collected, homogenized, and stored in TRIzol Reagent (Invitrogen, Carlsbad, CA, USA) at -80˚C. RNA was extracted according to the manufacturer's protocol. RNA was eluted in 25 μl of DNA/RNA free water and treated with DNAse (Life Technologies, Carlsbad, CA, USA). From this RNA, DENV and CHIKV loads were quantified using 4× TaqMan Fast Virus 1-Step Master Mix (Applied Biosystems, Foster City, CA, USA) in individual 10 μl reactions containing virus-specific primers and probes (Table 1). RT-qPCR reactions were run in a LightCycler 480 instrument (Roche Applied Science, Switzerland). The thermal cycling conditions were 50˚C for 5 min for reverse transcription, 95˚C for 10s for RT inactivation/denaturation followed by 50 amplification cycles of 95˚C for 3s, 60˚C for 30s, and 72˚C for 1s. Standard curves for both DENV and CHIKV were generated as described elsewhere [67,106] and contained a ~100 bp fragment of the 3'UTR of the DENV genome and a ~140 bp fragment of the CHIKV genome, respectively. The standard curve spanned from 10 to $10^7$ copies/reaction with a limit of detection of 100 copies for both viruses. The viral load in each sample was extrapolated from the standard curve as copies per mosquito.

## Estimation of viral load heritability

We tested for genetic variation and broad-sense heritability ($H^2$) for DENV and CHIKV loads across our family design. We estimated the parameters using a modified full-sib breeding design and the random effect linear model previously described [57,67]:

$$Z_{ij} = f_i + E_{ij} \tag{1}$$

$Z_{ij}$ is the trait value for the $j$th female from the $i$th family, $f_i$ is the random effect of the $i$th family, and $E_{ij}$ the unexplained error. To test if the genetic variation was significant, the family

**Table 1. Primers and probes for virus quantification.**

| GenBank ID | Virus | Direction | Sequence (5'- 3') |
|---|---|---|---|
| NC_001474.2 | DENV-2 | Forward | AAGGACTAGAGGTTAGAGGAGACCC |
| | | Reverse | CGTTCTGTGCCTGGAATGATG |
| | | Probe | FAM-AACAGCATATTGACGCTGGGAGAGACCAGA-BHQ1 |
| MT228632 | CHIKV | Forward | CACCCGAAGTAGCCCTGAATG |
| | | Reverse | TCCGAACATCTTTCCTCCCG |
| | | Probe | 5CY5-GAGAATAGCCCGCTGTCTAGATCCAC-3BHQ2 |

term was fitted as a random effect and the model (1) was compared to a reduced model without the family term. Loglikelihoods of both models were compared and twice the difference was tested against a chi-squared distribution with a single degree of freedom. Broad sense heritability was estimated as twice the family variance component ($\sigma_{family}$) divided by the total phenotypic variance ($\sigma_{family} + \sigma_{error}$), [107]. All models were constructed in SAS Studio version 3.8 (SAS Institute, Cary, NC, USA). Genetic correlations between DENV and CHIKV infected siblings were estimated using a bivariate version of model (1) and fitting unrestrictive covariance correlations at the family level (TYPE = UNR option) using the SAS Proc MIXED command. The significance of the genetic correlation was tested by loglikelihood ratio tests.

## RNA library preparation and sequencing

We selected four extreme families (Fig 1) from the high (families 59, 26, 81, and 12) and low (families 71, 1, 90, 16) ends of the CHIKV viral load distribution and pooled equal amounts of total body RNA from 4–5 mosquitoes per family. RNA concentrations were quantified using a Qubit 2.0 Fluorometer (ThermoFisher Scientific, Waltham, MA, USA), and RNA integrity was checked with a 4200 TapeStation (Agilent Technologies, Palo Alto, CA, USA) following manufacturer protocols. rRNA was depleted using a QIAGEN FastSelect rRNA HMR Kit (Qiagen, Hilden, Germany). We then used an NEBNext Ultra II RNA Library Preparation Kit for Illumina, following the manufacturer's recommendations (NEB, Ipswich, MA, USA), to prepare RNA sequencing libraries. Briefly, enriched RNAs were fragmented for 15 m at 94˚C. First-strand and second-strand cDNAs were subsequently synthesized, end-repaired, adenylated at 3'ends, and universal adapters were ligated to cDNA fragments followed by index addition and library enrichment with limited-cycle PCR. Sequencing libraries were validated using the Agilent Tapestation 4200 (Agilent Technologies, Palo Alto, CA, USA), and quantified using a Qubit 2.0 Fluorometer (ThermoFisher Scientific, Waltham, MA, USA) as well as by quantitative PCR (KAPA Biosystems, Wilmington, MA, USA). The sequencing libraries were multiplexed and clustered on one flowcell. After clustering, the flowcell was loaded on the Illumina HiSeq instrument according to the manufacturer's instructions. The samples were sequenced using a 2x150 Pair-End (PE) configuration. Raw sequence data (.bcl files) generated from Illumina HiSeq were converted into fastq files and de-multiplexed using Illumina bcl2fastq program version 2.20. One mismatch was allowed for index sequence identification.

## Data analysis

After demultiplexing, sequence data were checked for overall quality and yield. Sequence reads were then trimmed to remove adapter sequences and nucleotides with poor quality using Trimmomatic v.0.36 [108,109]. The trimmed reads were mapped to the reference genomes using the STAR aligner v.2.5.2b [109]. The STAR aligner is a splice-aware aligner that detects and incorporates splice junctions to help align the entire read sequences. BAM files were generated as a result of this step. Unique gene hit counts were calculated by using featureCounts from the Subread package v.1.5.2 [110]. Only unique reads within exon regions were counted. After extraction of gene hit counts, the gene hit counts table was used for downstream differential expression (DE) analysis. Using DESeq2 [111], we compared gene expression between groups. We used Wald tests to evaluate the significance and calculated log2 fold changes. Genes with adjusted $P$ values < 0.05 and absolute log2 fold changes >1 were called as differentially expressed genes for each comparison.

## Gene expression

In a subsequent experiment, so that different tissues and time points could be explored, mosquitoes were fed CHIKV, DENV, or mock (virus-free culture media) under the same conditions as for the heritability measures (as above, virus load $10^5$/ml), and then tissues (midgut, carcass, salivary glands) were dissected at 4-, 7-, and 14-days post-infection (dpi). Rather than demonstrating expression patterns across family groups, we sought to characterize the baseline and induced expression of the candidate genes. Tissues were dissected on ice in PBS and then homogenized in Trizol with a glass bead as above and as previously described [15]. RNA extraction was then carried out as above. Retrotranscription of RNA to cDNA and gene expression analysis was carried out on a LightCycler 480 instrument (Roche) using the Script One-step SYBR Green qRT-PCR (Quantabio, Beverly, MA, USA) according to the manufacturer's protocol. All CT values for AAEL004181 (Forward 5'- GCCATCGCCGCAACTT-CAGC -3', Reverse 5'- CACCCATGGCTCCCGATCCG -3') were normalized to the housekeeping *Ae. aegypti gene RpS17 or AAEL004175* (Forward 5'- TCCGTGGTATCTCCAT-CAAGCT-3', Reverse 5'-CACTTCCGGCACGTAGTTGTC-3') as per previous [112]. Primers for AAEL004181 were designed using Primer3 v 0.4.0 Gene expression ratios were obtained using the ΔΔCt method [113]. PCR amplification cycled 45 times at 95˚C for 3 sec and 60˚C for 30 sec, and the final cycle was followed by a melting curve analysis. Expression of RpS17 did not vary between virus-infected vs mock-infected control mosquitoes for any of the tissue x time point treatments examined (S1 Fig).

## Supporting information

**S1 Fig. RpS17 Ct values (mean and standard deviation) for DENV, CHIKV, and Mock-infected (fed virus-free culture media) mosquitoes.** Three tissues, midgut (MG), carcass (Car), and salivary glands (SG), were dissected on three different days post-infection (dpi). (PDF)

## Acknowledgments

The authors would like to thank Michael Cannon for his assistance with mosquito rearing.

## Author Contributions

**Conceptualization:** Mario Novelo, Stephen F. Chenoweth, Elizabeth A. McGraw.

**Data curation:** Elizabeth A. McGraw.

**Formal analysis:** Mario Novelo, Hillery C. Metz, Scott L. Allen, Stephen F. Chenoweth, Elizabeth A. McGraw.

**Funding acquisition:** Elizabeth A. McGraw.

**Investigation:** Mario Novelo, Heverton LC Dutra, Matthew J. Jones, Leah T. Sigle.

**Methodology:** Heverton LC Dutra, Matthew J. Jones, Leah T. Sigle.

**Project administration:** Elizabeth A. McGraw.

**Supervision:** Francesca D. Frentiu, Stephen F. Chenoweth, Elizabeth A. McGraw.

**Writing – original draft:** Mario Novelo, Hillery C. Metz, Francesca D. Frentiu, Scott L. Allen, Stephen F. Chenoweth, Elizabeth A. McGraw.

**Writing – review & editing:** Mario Novelo, Heverton LC Dutra, Hillery C. Metz, Matthew J. Jones, Leah T. Sigle, Francesca D. Frentiu, Scott L. Allen, Stephen F. Chenoweth, Elizabeth A. McGraw.

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
