## [Decision Letter · Decision Letter 0]

15 Jan 2023

Dear Prof McGraw,

Thank you very much for submitting your manuscript "Dengue and chikungunya virus loads in the mosquito Aedes aegypti are determined by distinct genetic architectures" for consideration at PLOS Pathogens. As with all papers reviewed by the journal, your manuscript was reviewed by members of the editorial board and by several independent reviewers. The reviewers appreciated the attention to an important topic. 

I am returning your manuscript with three reviews. The reviewers came to different conclusions about the details and analysis of the results, as you will see. After reading the reviews and looking at the manuscript, I have decided that the major revisions requested by reviewer 1 can be addressed without new experiments but do require adding more complete data (eg.: RNA-seq and qPCR). There are also a few remaining minor revisions that need to be addressed to prepare the manuscript for publication.

If all the following items are addressed, I hope to be able to make a final decision without sending the manuscript out for a second round of review. 

Please pay particular attention to the following reviewer suggestions and give them due consideration.

x  Please make your RNA-seq datasets publicly available by depositing your datasets in a publicly accessible repository

x  Please add a table or a graph with the Ct values for RpS17 and its standard curve  to assess the consistency of this housekeeping gene among tissues and mosquitoes for the normalization of the qPCR data.

x  Specify how many mosquitoes were used for the tissue-specific expression of AEL004181 experiment and from what mosquito families.

x  Make sure to include the Table 1 (primers and probes), which is currently missing.

x Please explain what “High and Low CHKV families” are in the introduction (even if it is explained in the Results section).

x Clarify the description of the gene LOC110676965 in  the results section. It is first described as encoding a Zinc finger containing protein (line243) but is later described as a long non coding RNA (lncRNA, line258).

The following is a more minor point:

x Consider adding a schematic diagram to summarize the main experimental design with the breeding approach and infection experiments.

Sincerely,

Michel Tassetto

Guest Editor

PLOS Pathogens

Raul Andino

Section Editor

PLOS Pathogens

Kasturi Haldar

Editor-in-Chief

PLOS Pathogens

orcid.org/0000-0001-5065-158X

Michael Malim

Editor-in-Chief

PLOS Pathogens

orcid.org/0000-0002-7699-2064

Reviewer Comments (if any, and for reference):

Reviewer's Responses to Questions

**Part I - Summary**

Reviewer #1: In the work by Novelo et al, the authors estimate the contribution of mosquito genetic variation to viral loads of both DENV and CHIKV in Aedes aegytpi, in full-sibling design, suggesting that these respond to the two viruses using distinct genetic mechanisms.  Using RNAseq, the authors, identified only two loci that consistently differentiated High and Low CHIKV families. The study is interesting from the point of view of vector competence and genetic factors associated with viral loads of two major medically important viruses transmitted by mosquito vectors.

Although very interesting data are presented, and the experimental model is adequate, there is a high variability shown in viral loads in both viruses in the various groups of families (Figure 1), families with lower and higher viral loads have been taken expecting to see differences and only differences in expression in two genes have been observed (the authors should include the supplementary tables with the complete data of the RNAseq and I consider It is important to take at least two more families with intermediate viral loads and observe if these differences are actually due to the effect of the viral load and not to the variability itself of the population of mosquitoes analyzed), by the analysis carried out to become important for vector competence tests.

It should be noted that the authors should focus on and specify both the introduction and the discussion on the problem to be treated and the results obtained. From line 100 to line 126 in the introduction, although interesting data for the general public, they seem irrelevant to the topic addressed in this work and could channel readers to a review of the topic. Lines 139-154 should specify the information that has been collected on mosquitoes and leave out data on drosophila. The discussion to take into consideration ISVs and MSVi is interesting but it also rambles and falls into speculations that do not entail support of the data obtained so I suggest limiting these points and focusing the discussion of what has been obtained in figure 1 and figure 3, taking into account the high variability in viral loads that have been observed despite having a family model of siblings and the expression of AAEL004181 in the tissues evaluated (because it is an SGS is expected to be a greater expression of this in glands? What repercussions would it have for viral transmission, will it be relevant to prove its presence in shorter times taking into account the differential extrinsic incubation period in both viruses?). In summary, the introduction and discussion are very long and the meaning and relevance of the data obtained are lost.

Reviewer #2: This study represents a modified full-sib breeding design to study the contribution of the

Aedes aegypti’s genetic variation to vector competence of dengue and chikungunya viruses. Remarkably, they found no genetic correlation between DENV and CHIKV loads between siblings, suggesting independent genetic responses to these two viruses in different families. The authors additionally measured gene expression among high and low viral load families and found differentiated loci at different time points following infection. This student is well-executed and well-written. The results will help advance the field’s understanding of vector-virus interactions in these two globally important arboviruses. The methods and results are clear and I only have minor suggestions for improvements.

Reviewer #3: This manuscript provide a good finding and latest knowledge on the interaction between vector and virus load which is one of significant parts in control approaches for mosquito-borne diseases. It also been well written and the data has been scientifically analysis. The findings have been well explained in discussion section and understandable for critical thinking level. However, I just not sure the arrangement of the subtitle in this manuscript, it should be the Method section before the Result section then ended with discussion and conclusion. Overall, I am really recommending accept for publication due to the high quality of the research has been conducted and the authors did a great transformation it into knowledgeable manuscript for expert referred in future.

**Part II – Major Issues: Key Experiments Required for Acceptance**

Reviewer #1: I believe that the authors should include in supplementary data the levels of expression obtained from the RNAseq. The authors did not include Table 1. I consider that they should include in supplementary data the normalization graph of the RpS17 gene in each of the mosquito tissues evaluated in order to verify that this gene is good housekeeping in the tissues evaluated.

Figure 1, explains in detail in the experiments why only 37 families were taken if they could have been more considering that I have understood that 600 mosquitoes were placed individually, and according to details of the methods have been used only those that laid more than 60 eggs. Is that why only 37 have remained?

It has caused me confusion, in the methods it is indicated that the families would consist of at least 8 females for DENV and 8 for CHIK but in figure 1 an N of 3-5 per family is mentioned, what has happened to the rest of the mosquitoes in the family? Will this low number of mosquitoes considered by each family influence the viral load? Obviously, it has influenced the deviation of each of the families for example in the DENV 85 family where one of the mosquitoes that integrate it has been infected with too much and two others have not, with this argument valuable data has been left aside for the RNAseq?

It is recommended that the authors incorporate these details into the methodology and clear these doubts.

It is requested improve the quality of Figure 3.

Reviewer #2: (No Response)

Reviewer #3: None

**Part III – Minor Issues: Editorial and Data Presentation Modifications**

Reviewer #1: I also strongly recommend including a figure with the experimental model to make it more understandable, this model should include the information of the mock mosquitoes that were used for the relative expression tests by qPCR since they have not been specified in the methods.

Reviewer #2: Specific comments

Line 68: Should be ‘Arthropod-borne’

Ln. 361. Speaking of MSVs, did the authors (or anyone) screen these populations of Ae. aegypti from Monterrey for ISVs? The presence of MSFs that are flaviviruses or alphaviruses could have had relevance for the observations in the present study.

Ln. 419. You could clarify here that you mean individualized gene editing for controlling different pathogens would be needed.

Ln 429. I don’t understand how an F3 line of Ae. aegypti collected in Monterrey, Mexico was then reared for another three generations to adapt to colony feeding. Does this mean someone in Monterrey started this colony in Mexico for three generations, and then the F3s were shipped to Penn State? It would help to clarify this history. Is there a reference for the collection of Ae. aegypti eggs from traps (I assume these are ovitraps) in Monterrey that can be cited?

Ln. 480. If specifying where CHIKV challenges were conducted, would be good to clarify where the DENV challenged occurred.

Ln. 555. Would be good to clarify that the Trizol was part of the RNA extraction step as described above.

Reviewer #3: The subtopics arrangement such as the method section should be before result section.

PLOS authors have the option to publish the peer review history of their article (what does this mean?). If published, this will include your full peer review and any attached files.

Reviewer #1: **Yes: **Jorge Cime-Castillo

Reviewer #2: No

Reviewer #3: No

Figure Files:

Data Requirements:

Reproducibility:

References:

---

## [Editor Report · Decision Letter 1]

19 Mar 2023

Dear Prof McGraw,

We are pleased to inform you that your manuscript 'Dengue and chikungunya virus loads in the mosquito Aedes aegypti are determined by distinct genetic architectures' has been provisionally accepted for publication in PLOS Pathogens.

Best regards,

Raul Andino

Academic Editor

PLOS Pathogens

Raul Andino

Academic Editor

PLOS Pathogens

Kasturi Haldar

Editor-in-Chief

PLOS Pathogens

orcid.org/0000-0001-5065-158X

Michael Malim

Editor-in-Chief

PLOS Pathogens

orcid.org/0000-0002-7699-2064
---

## [Editor Report · Acceptance letter]

5 Apr 2023

Dear Prof McGraw,

We are delighted to inform you that your manuscript, "Dengue and chikungunya virus loads in the mosquito *Aedes aegypti* are determined by distinct genetic architectures," has been formally accepted for publication in PLOS Pathogens.

Best regards,

Kasturi Haldar

Editor-in-Chief

PLOS Pathogens

orcid.org/0000-0001-5065-158X

Michael Malim

Editor-in-Chief

PLOS Pathogens

orcid.org/0000-0002-7699-2064